# Beauty Beware: Associations between Perceptions of Harm and Safer Hair-Product-Purchasing Behaviors in a Cross-Sectional Study of Adults Affiliated with a University in the Northeast

**DOI:** 10.3390/ijerph20237129

**Published:** 2023-11-30

**Authors:** Cathryn E. Payne, Amber Rockson, Adiba Ashrafi, Jasmine A. McDonald, Traci N. Bethea, Emily S. Barrett, Adana A. M. Llanos

**Affiliations:** 1Department of Epidemiology, Mailman School of Public Health, Columbia University Irving Medical Center, New York, NY 10032, USA; cep2165@caa.columbia.edu (C.E.P.); ar4571@cumc.columbia.edu (A.R.);; 2Herbert Irving Comprehensive Cancer Center, Columbia University Irving Medical Center, New York, NY 10032, USA; 3Lombardi Comprehensive Cancer Center, Georgetown University, Washington, DC 20057, USA; tb988@georgetown.edu; 4Department of Biostatistics & Epidemiology, Rutgers School of Public Health, Piscataway, NJ 08854, USA; esb104@eohsi.rutgers.edu; 5Environmental and Occupational Health Sciences Institute, Rutgers University, Piscataway, NJ 08854, USA

**Keywords:** personal care products, hair products, perceptions of harm, adults, endocrine-disrupting chemicals

## Abstract

(1) Background: Widespread personal care product (PCP) use can expose individuals to endocrine-disrupting chemicals (EDCs) associated with adverse health outcomes. This study investigated the association between harm perceptions and hair-product-purchasing behaviors in adults enrolled in a cross-sectional study. (2) Methods: Respondents rated their agreement with five PCP-related harm statements using a five-point Likert scale. Multivariable-adjusted logistic regression models were used to examine the associations between harm perceptions with hair-product-purchasing behaviors and hair product use (i.e., number of products used). (3) Results: Among 567 respondents (non-Hispanic White, 54.9%; non-Hispanic Black, 9.5%; Hispanic/Latinx, 10.1%; Asian American/Pacific Islander, 20.1%; and multiracial/other, 5.5%), stronger harm perceptions around PCP use were associated with potentially “safer” hair-product-purchasing behaviors. Respondents who strongly agreed that consumers should be concerned about the health effects of PCPs had more than fourfold increased odds of always/usually using healthy product apps (OR 4.10, 95% CI: 2.04–8.26); reading ingredient labels (OR 4.53, 95% CI: 2.99–6.87); and looking for natural, non-toxic, or eco-friendly product labels (OR 4.53, 95% CI: 2.99–6.88) when buying hair products. (4) Conclusions: Promoting environmental health literacy and raising awareness of potential PCP use-related harms might encourage healthier hair product use behaviors.

## 1. Introduction

Daily use of personal care products (PCPs), including skincare, hair, beauty, and menstrual products (e.g., tampons, sanitary pads, menstrual cups, washes, and wipes) is ubiquitous. These products often contain endocrine-disrupting chemicals (EDCs) that can alter the normal mechanisms of the endocrine system responsible for human reproduction, growth, and development [1]. EDCs of concern include parabens, phthalates, and benzophenone, which are added to PCPs as preservatives to prevent the growth of microorganisms, make products durable, or enhance colors and scents [2,3]. A growing body of literature suggests that individuals who use certain PCPs, particularly hair dye and relaxers, are at an increased risk for a range of adverse reproductive health outcomes, including younger age at menarche and ovarian, breast, or uterine cancer [4,5,6,7,8,9,10,11,12]. Due to these findings, there are concerns about how exposure to EDCs contained in these products may impact health, particularly given frequent, long-term use. The Food and Drug Administration (FDA) has minimal regulation over these PCPs, which often leaves the burden of risk reduction on consumers [13]. However, individuals can reduce their chemical exposure by using products without EDCs, reducing the number of products used, or avoiding certain products altogether. Understanding why individuals select products they purchase for use may be useful for informing strategies to promote healthier product use as a means of reducing chemical exposures from PCP use.

According to one study [14], individuals who read ingredient labels to avoid certain chemicals or who avoided certain products altogether had lower urinary concentrations of parabens, triclosan, and benzophenone-3. In another study [15], which examined the impact of an intervention on adolescent product use, researchers found that choosing products labeled “free of phthalates, parabens, triclosan, and benzophenone-3” was associated with lower urinary concentrations of these EDCs. Moreover, a separate study showed that using paraben-free and phthalate-free products over 28 days was associated with both reduced urinary concentrations of these EDCs and a reversal of cancer-associated phenotypes at the cellular and molecular level in healthy breast tissues (namely, transcriptional shifts in the expression of known cancer-associated genes and shifts toward the normalization of estradiol-modulated pathways) [16]. While these studies identified associations between product use and urinary concentrations of EDCs or cancer-associated phenotypes, they did not examine what factors motivate these safer consumer behaviors.

Numerous healthy product smartphone applications and online resources (i.e., Environmental Working Group’s (EWG) Healthy Living app or Skin Deep^®^ online cosmetics database, Silent Spring Institute’s Detox Me app, and the Clearya internet browser extension and app) are freely available to consumers who wish to obtain more information about the chemicals found in PCPs prior to purchase. In a diverse sample of women, researchers documented that nearly half of the respondents bought products based on their ingredients, as opposed to other deciding factors (e.g., price, brand, effectiveness), but did not go on to research what motivated participants to examine the ingredient label [17]. Existing literature also suggests that using more PCPs daily—which increases the number of chemicals individuals are exposed to—is associated with a higher body burden of EDCs [14,15,18,19,20,21,22]. Taken together, these findings suggest that certain behavioral modifications might effectively reduce exposures to harmful chemicals through PCP use, though limited research has examined how factors, particularly attitudes and perceptions, motivate these behaviors.

Some evidence suggests that knowledge and perceptions toward environmental chemical exposures may influence PCP purchasing and use. In studies that examined perceptions of harm, 51% to 94% of respondents expressed concerns about the dangers or health effects of everyday or environmental chemicals [23,24,25]. In a study that examined knowledge, attitudes, and behaviors around chemical exposures, researchers found that older age and healthcare professions were associated with perceiving chemicals in the environment as dangerous [24]. Furthermore, women who agreed that chemicals were unsafe were more inclined to try to limit their exposure while purchasing products [24]. These findings notwithstanding, the relationship between perceptions of harm and “safer” product purchasing behaviors has been inadequately examined.

To better understand these relationships, we assessed whether the perceptions of harm around PCP and hair product use are associated with hair-product-purchasing behaviors and the number of hair products used overall in a cross-sectional study of adults residing in the Northeastern United States (US). We focused the current analysis on hair product behaviors given our recent findings showing significant gender and racial and ethnic differences in hair product usage [26]. Unsurprisingly, we found that women use more hair products than men [26]. We also found that relative to non-Hispanic White (NHW) women, non-Hispanic Black (NHB) women use more hair products, while Asian American/Pacific Islander women use fewer hair products [26]. Further, given the strong ties between hair styling and identity, the variation in common hair-styling practices depending on hair type, and differences in the toxicity of hair product types, we were interested in exploring the role of environmental health literacy [27] based on perceptions around PCP use in the context of hair product use specifically. We hypothesized that greater perceptions of harm around PCP use are associated with increased odds of “safer” hair-product-purchasing behaviors, namely, using a healthy product app or website; reading product ingredient lists; and seeking products labeled as natural, non-toxic, or eco-friendly when purchasing hair products to use at home. Additionally, we hypothesized that greater perceptions of harm around PCP use are associated with using fewer hair products overall.

## 2. Materials and Methods

### 2.1. Data Source and Study Participants

This study was a secondary analysis of cross-sectional data collected from September to October 2019 as part of a larger, questionnaire-based study on PCP use and perceptions. As previously described [26], an electronic survey was disseminated to members of a college campus in New Jersey (Rutgers University). The questionnaire ascertained sociodemographic characteristics, PCP purchasing and use, and perceptions of harm around using PCPs. Respondents included university students, staff, and faculty (*n* = 657). In this analysis, we excluded individuals who did not return a completed questionnaire (*n* = 79), those with missing data on age, income, or education (*n* = 9), and those who self-identified as a gender other than female or male (due to the small sample size: non-binary (*n* = 1) or genderqueer (*n* = 1) (Figure 1)). Informed consent and individual questionnaire responses were acquired using Qualtrics. This study was approved by the Institutional Review Boards of Rutgers University and Columbia University Irving Medical Center.

### 2.2. Independent Variables

Sociodemographic characteristics queried included age, gender identity, race, ethnicity, educational attainment, annual household income, marital status, and role at the university. Gender identity was categorized as female or male; race and ethnicity were categorized as non-Hispanic White (NHW), non-Hispanic Black (NHB), Hispanic/Latinx, Asian American/Pacific Islander (AAPI), or multiracial/other; education was categorized as high school diploma or equivalent, some college, bachelor’s degree, master’s degree, or doctoral degree; annual household income was categorized as <USD 50,000, USD 50,000–99,999, USD 100,000–149,000, or ≥USD 150,000; marital status was reported as married, divorced, widowed, separated, or single/never married and then dichotomized as married or unmarried in subsequent analyses; and university role was categorized as undergraduate student, graduate student, staff, faculty, or other.

Perceptions around PCPs and PCP use were assessed by asking the participants to indicate their degree of agreement with five statements, each with a 5-point Likert scale (strongly agree, agree, neither agree nor disagree, disagree, or strongly disagree): (1) “The [PCPs] I use affect my health,” (2) “Consumers should be concerned about the health effects of [PCPs],” (3) “There is no reason to worry about the health effects from chemicals that might be in [PCPs],” (4) “Overall, the benefits of using [PCPs] outweigh any risk from exposure to toxic chemicals that might be in these products,” and (5) “The Food and Drug Administration (FDA) and other government agencies do a good job of regulating [PCPs] to ensure they are safe for consumers.” It should be noted that these study questions were developed by a team of experts on chemical exposures in PCPs because no validated questionnaires exist to date examining the perceptions of harm toward PCPs specifically. Based on the distribution of responses in the overall study sample, responses to each statement were dichotomized to reflect greater perceptions of harm/concern versus lower perceptions of harm/concern toward all PCPs (e.g., “strongly agree” versus all other response options for statements 1 and 2 and “strongly disagree” versus all other response options for statements 3, 4, and 5).

### 2.3. Dependent Variables

The main outcomes of interest—hair-product-purchasing behaviors—were assessed by asking participants to indicate how often they practiced three behaviors when buying a hair product to use at home, each with 5-point Likert scale options (never, rarely, sometimes, usually, or always): (1) use a healthy product app or website (e.g., EWG and Skin Deep^®^); (2) read the ingredients on the label; or (3) look for labels indicating the product was made with natural, non-toxic, or eco-friendly ingredients. Based on the distribution of responses in the overall study sample, the responses were dichotomized to reflect always/usually practicing the behavior (hereafter referred to as “usually”) versus all other response options.

Information about hair products used in the last 24–48 h was collected by asking the participants to list what hair products they used, including shampoo, conditioner, detangler, dandruff/scalp treatment products, hair-styling products, hair/scalp treatments (e.g., hair spray, hair oil, hair gel, pomade, styling gel/lotion, styling mousse/foam, hair serum, hair food, natural/essential oils), and hair loss treatment products. The total count of hair products used by participants was based on the overall number of hair products participants reported using in the last 24–48 h. Hair product use was dichotomized based on the median number of hair products (≥3 vs. <3) reported by the study participants in the analytic sample, which remained the same among all participants and when looking at females in our gender-stratified models.

### 2.4. Statistical Analyses

We used descriptive statistics (i.e., frequencies, proportions, means, and standard deviations (SDs)) to summarize the sociodemographic characteristics of the study sample. For the main analysis examining both male and female participants, multivariable logistic regression models were used to assess the relationships of each of the five perceptions of harm with engagement in three behaviors when purchasing hair products and the total number of hair products used, adjusting for gender, age, income, marital status, and race and ethnicity, which were selected a priori based on existing literature depicting the differences in use across these characteristics [6,14,17,23,24]. Odds ratios (ORs) and 95% confidence intervals (CIs) were estimated from these logistic regression models. We sought to also assess these associations in gender-stratified models; however, due to limited power for the total number of hair products used among male participants (i.e., 3 male participants reported using >3 hair products), we limited further analysis of the associations between perceptions of harm and hair product use to female participants. All statistical analyses were performed in SAS version 9.4 (SAS Institute, Cary, NC).

## 3. Results

### 3.1. Sample Characteristics and Distributions of Perceptions of Harm and Hair-Product-Purchasing Behaviors

Data on 567 female and male respondents were analyzed (Figure 1). On average, participants were aged 37.0 ± 15.6 years (range: 18 to 79 years). Most identified as female (85.5%), were NHW (54.9%), had an undergraduate degree (74.2%), had household income > USD 50,000 (84.2%), were unmarried (61.9%), and were staff (40.2%) at the university (Table 1). The distributions of perceptions of harm and hair-product-purchasing behaviors are presented in Table 2. Less than one-fifth of the sample (13.6%) strongly agreed that PCPs affect their health. Over one-quarter (27.0%) strongly agreed that consumers should be concerned about the health effects of PCPs, and over one-third (33.9%) of participants strongly disagreed that there was no reason to worry about the health effects of chemicals in PCPs. Furthermore, 15.2% of respondents strongly disagreed that the benefits of using PCPs outweighed any risks from chemicals that might be in PCPs. Less than one-fifth (16.1%) strongly disagreed that the FDA and other governmental agencies did a good job of regulating PCPs to ensure they are safe for consumers. Only 7.0% of participants indicated that they usually used a healthy product app when purchasing hair products, while 38.3% indicated that they read the ingredients, and 37.4% indicated that they usually looked for labels indicating the product was made with natural, non-toxic, or eco-friendly ingredients.

### 3.2. Associations between Perceptions of Harm and Hair-Product-Purchasing Behaviors

In the multivariable logistic regression models, perceptions of harm were associated with “safer” purchasing behaviors (Figure 2). When compared with respondents who did not strongly agree, those who strongly agreed that PCPs affected their health had more than twofold increased odds of reporting that they usually used a healthy product app (OR: 2.46, 95% CI: 1.10–5.53); usually read ingredient labels (OR: 3.60, 95% CI: 2.13–6.08); and usually looked for labels indicating products are natural, non-toxic, or eco-friendly (OR: 3.63, 95% CI: 2.15–6.13) when buying hair products to use at home. Relative to those who did not strongly agree, those who strongly agreed that consumers should be concerned about the health effects of PCPs had more than fourfold increased odds of reporting that they usually used a healthy product app or website when buying a hair product to use (OR 4.10, 95% CI: 2.04–8.26); usually read ingredient labels (OR: 4.53, 95% CI: 2.99–6.87); and usually looked for labels indicating products are natural, non-toxic, or eco-friendly (OR: 4.53, 95% CI: 2.99–6.88) when buying hair products to use at home. Similarly, compared with those who did not strongly disagree, those who strongly disagreed that there was no reason to worry about the health effects from chemicals that might be in PCPs had more than threefold increased odds of reporting that they usually used a healthy product app or website (OR 3.33, 95% CI: 1.64–6.76); usually read ingredient labels (OR: 3.03, 95% CI: 2.07–4.45), and usually looked for labels indicating products are natural, non-toxic, or eco-friendly (OR: 3.09, 95% CI: 2.10–4.54) when buying a hair product to use at home. In summary, those who expressed concerns about the health effects of chemicals that might be in PCPs had increased odds of reporting that they followed “safer” purchasing behaviors.

Participants who indicated that the benefits of using PCPs did not outweigh any risks from exposure to toxic chemicals that might be in products had twofold increased odds of reporting that they usually used a healthy product app (OR: 2.65, 95% CI: 1.24–5.66); usually read ingredient labels (OR: 3.55, 95% CI: 2.15–5.86); and usually looked for labels indicating products are natural, non-toxic, or eco-friendly (OR: 3.56, 95% CI: 2.16–5.88) when buying hair products to use at home compared with those who did not strongly disagree. Finally, those with perceptions indicating that the FDA and other governmental agencies did not do a good job of regulating PCPs to ensure they are safe for consumers had increased odds of usually using a healthy product app or website (OR: 3.24, 95% CI: 1.49–7.04) when buying hair products to use at home. As shown in Figure 3, the perceptions of harm were not significantly associated with the number of hair products used in the past 24–48 h.

These findings were generally consistent in our gender-stratified models. Among the female participants (Appendix A), the associations were nearly identical and remained statistically significant, except for the associations with the FDA and other governmental agencies, which were slightly attenuated. The models among male participants yielded less reliable estimates of the associations between the perceptions of harm and hair-product-purchasing behaviors (Appendix A), although the unadjusted estimates were relatively consistent in terms of the direction of the associations reported in our primary analysis, albeit imprecise.

## 4. Discussion

This research, using data from a subset of adults who participated in a questionnaire-based study conducted on a college campus in the Northeastern US, adds to the limited literature on the associations between the perceptions of harm associated with PCPs in relation to PCP use behaviors. This study also represents a relatively large sample, which was inclusive of >40% racial and ethnic minorities and almost 15% who identified as male, which is a population that is often excluded from studies related to PCP use. Our findings support the hypothesis that perceptions suggestive of strong agreement that PCPs are harmful were often associated with increased odds of “safer” purchasing behaviors, as measured by the reported use of healthy product apps; reading the ingredient label; or looking for labels indicating the product is natural, non-toxic, or eco-friendly when purchasing hair products for home use. Interestingly, greater perceptions of harm were not significantly associated with the number of hair products participants typically used. This might indicate that the combination of greater environmental health literacy and safer purchasing behaviors may not influence the number of hair products individuals use. Alternatively, it may depict that the products participants are using could be less toxic (i.e., contain fewer chemicals of concern); this warrants further consideration.

Our findings illustrate that strong perceptions of harm were associated with usually looking for labels indicating the product was made with natural, non-toxic, or eco-friendly ingredients when buying a hair product to use at home. This result is consistent with findings from another study showing that individuals who agreed that chemicals were dangerous had increased odds of trying to purchase “eco-friendly” or “chemical-free” PCPs [24]. However, “greenwashing”—which is falsely advertising products to make them appear environmentally sound—is made possible by a lack of green standards, which makes this a less effective approach [28,29]. More recently, “clean washing” was coined to describe falsely advertising products to make them appear safer or healthier [30,31]. These terms illustrate one of the many barriers faced by consumers. Unequal access to potentially safer products for women of color and the perception that these products are less effective are additional barriers that inhibit the use of safer, cleaner, and less toxic products [29,32,33,34]. In a study examining Black women’s perceived barriers to and benefits of using eco-products, participants listed multiple barriers when purchasing these products, which include lack of access to the products in the community, cost, and racism/colorism [29].

In our study sample, consumers with perceptions suggestive of strong agreement that PCPs are consistently harmful had increased odds of using healthy product applications. To our knowledge, no other study has examined the relationship between perceptions of harm and the use of a healthy product app or website. Smartphone apps and online resources are useful resources for consumers to research PCPs and other products they use in their homes to learn more about their chemical ingredients and potential health hazards [35]. As such, using these resources can empower consumers to make more informed decisions when purchasing products. While each of these apps has a growing number of products listed, they are not exhaustive of all products on the market. There is a need for inclusive resources since consumer groups differ in their needs related to PCP use and companies in the PCP industry have a history of engaging in product development and advertising practices that vary by consumer group. One project, namely, the Campaign for Safe Cosmetics’ Non-Toxic Black Beauty Project [36], provides a list of Black-owned beauty brands with safer product lines available for purchase. Additionally, resources containing searchable information on chemicals of concern could be another opportunity to empower consumers to make more informed decisions when purchasing PCPs, thereby mitigating potentially negative health outcomes from using these products. It should also be noted that no research has evaluated the relationship between using a healthy product application and the levels of EDCs detectable in the body. Additionally, it is unknown whether using a healthy product application is an effective approach to reducing urinary concentrations of certain EDCs when compared with reading the ingredient label. Nonetheless, the findings of the current study highlight how perceptions of harm influence purchasing behaviors and underscore the importance of accessible tools consumers can use to make more informed choices about the products they purchase, which may help to minimize their risk of exposure to EDCs.

In this study, 38% of participants indicated they usually read ingredients on product labels when buying a hair product, and strong perceptions of harm related to chemicals in PCPs were associated with this behavior. There are concerns regarding whether reading the ingredient lists on product labels actually reduces the risk of chemical exposure, as existing research indicates that when testing PCPs, many chemicals were detected that were not explicitly listed on the product label (predominantly cyclosiloxanes) [37,38]. Moreover, studies often examine the harmfulness of single chemical ingredients without examining the effect of chemical mixtures, which, in some cases, can underestimate the health effects, further reducing the efficacy of reading an ingredient label to reduce risks [39]. Differences in toxic chemical ingredients in PCPs are partly due to manufacturer’s choice and federal agencies’ enforcement of consumer product regulations, which contribute to differences in PCP safety [28,40,41]. For example, while manufacturers of cosmetics and other PCPs are required to list their ingredients on product labels, the FDA explicitly states that fragrance or flavor can be listed simply as such without the disclosure of the specific chemicals used to create the fragrance or flavor [40]. This reduces the transparency of product labeling, leaving consumers unaware of what is actually in their products. The chemicals that are added to fragranced products are considerable; some researchers found that when testing fragranced products, many chemicals were found that were not explicitly listed on the product label [37,38]. Other studies documented a positive association between the use of fragranced products and urinary concentrations of several phthalate metabolites [19,22,42]. The Modernization of Cosmetics Regulation Act (MoCRA) of 2022 now grants additional regulatory power to the FDA in this regard [43,44].

Overall, increased screening standards from regulatory agencies, like the FDA, are needed to mitigate the presence of harmful chemicals in PCPs and provide greater ingredient transparency on product labels. This action could provide consumers with more accurate and complete ingredient information, in turn reducing misinformation and helping to empower consumers. With current regulations, reading the ingredient label would not represent an exhaustive approach to identifying potentially harmful chemicals in products and choosing to avoid products that contain them as a means to reduce exposure. As such, additional regulations are needed to protect consumers. Although this study focused on individual behaviors, the onus should not be on consumers to differentiate environmentally sound products from greenwashing and other marketing tactics.

Notably, our findings do not support the hypothesis that greater perceptions of harm are associated with using fewer hair products. To our knowledge, this was the first study to examine the relationship between perceptions of harm and the total number of hair products used overall during a 24–48 h period. The lack of associations between perceptions of harm and total hair product count may indicate that greater perceptions of harm and exhibiting “safer” purchasing behaviors did not translate to the number of hair products individuals use, although our questionnaire only asked participants about PCP use within the last 24–48 h, which may not have been the optimal period to examine. Alternatively, it may be related to limited consumer awareness around the cumulative burden of the chemicals in PCPs. Research showed that the total product count is associated with higher levels of urinary chemical concentrations, particularly MEP [20,21,22]. Consumers may be unaware of these chemicals and their risks, as one study that examined participants’ knowledge of chemicals found that only 44% of the 871 women in their study had ever heard of phthalates [24]. One community-based participatory research study illustrated that targeted interventions focusing on chemical exposures through cosmetics are an effective measure to increase environmental health literacy as a means to promote healthier PCP behaviors [45,46]. However, one study noted that using PCPs labeled as free of phthalates, parabens, triclosan, and BP-3 can effectively reduce urinary EDC concentrations [15]. Therefore, it is possible that consumers—and respondents in the current study—might prefer to use cleaner, less toxic PCPs, but not necessarily to use fewer products overall. This may provide consumers with another strategy for maintaining a lower burden of exposure to potentially harmful chemicals without reducing the number of products they use. Further research is needed to test this hypothesis. Likewise, future analysis of the causal associations of perceptions of harm, total product use, and urinary biomarkers are warranted to fully understand the impact of perceptions of harm on PCP use behaviors and the subsequent impact of these behaviors on specific health outcomes, particularly among lower SES groups, who may not have access to “cleaner” and/or “safer” PCPs.

This study had several strengths, including a relatively diverse sample in terms of age, occupation, and income, although less diverse in terms of race and ethnicity. We also observed variations in numerous measures of perceptions of harm around the use of PCPs, which yielded novel results linking perceptions of harm from chemicals in PCPs with a few “safer” purchasing behaviors. This study also had several limitations that should be considered. First, this analysis focused on behaviors around hair-product-purchasing and did not consider other product categories assessed in the questionnaire (e.g., skin, beauty, and menstrual products), which also contain EDCs. Additionally, we were unable to perform race-and-ethnicity-stratified analyses due to sample size limitations. Furthermore, conducting this study on a college campus—where the annual estimated household income and educational attainment are higher than the general US population—may have limited the generalizability of these findings. We also did not examine whether other criteria, such as product brand preferences, perceived quality, word-of-mouth/friends’ recommendations, or social desirability, played a role in how consumers made choices around PCP purchasing [17,47]. The possibility remains that other unmeasured motivators or barriers may have influenced the relationship between perceptions of harm and “safer” product behaviors [17,29,47,48,49,50]. Alternatively, self-reported purchasing behaviors could suffer from social desirability bias, particularly among participants with an awareness that chemicals in these products may be harmful. Finally, because both the study exposure and outcome were measured on the same survey, there is a potential for dependent misclassification. Thus, this could have resulted in an unpredictable impact on our reported odds ratios.

## 5. Conclusions

Despite these limitations, our study’s findings illustrate that perceptions of harm from chemicals in PCPs were associated with how people chose the hair products they used at home. These findings might be useful for developing strategies to promote healthier PCP use as a means of reducing chemical exposures from these products. When purchasing products, consumers can attempt to reduce their risk of exposure to EDCs in PCPs through certain behavioral modifications. Educational interventions are another public health strategy that can be used to empower consumers to make informed decisions and reduce their risk of chemical exposure. Future research should examine the effectiveness of educational interventions to promote “safer” PCP use and learn about the barriers faced by consumers attempting to purchase products without harmful chemicals. And lastly, it is also important to consider that tighter regulation of potentially harmful chemicals is another strategy for mitigating exposure risk—one that takes the onus off the consumer.

## Figures and Tables

**Figure 1 ijerph-20-07129-f001:**
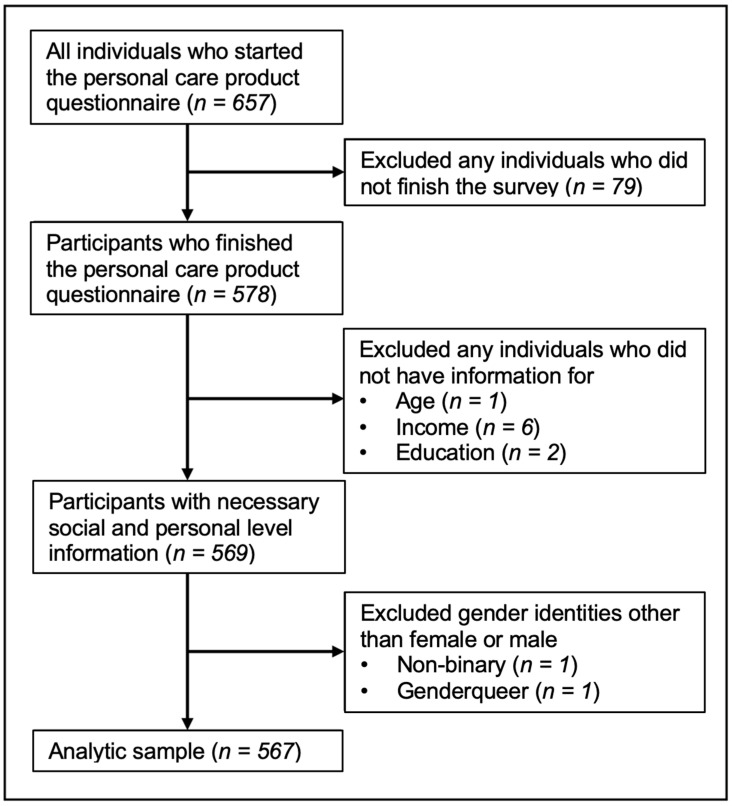
Flow diagram of participants based on exclusion criteria.

**Figure 2 ijerph-20-07129-f002:**
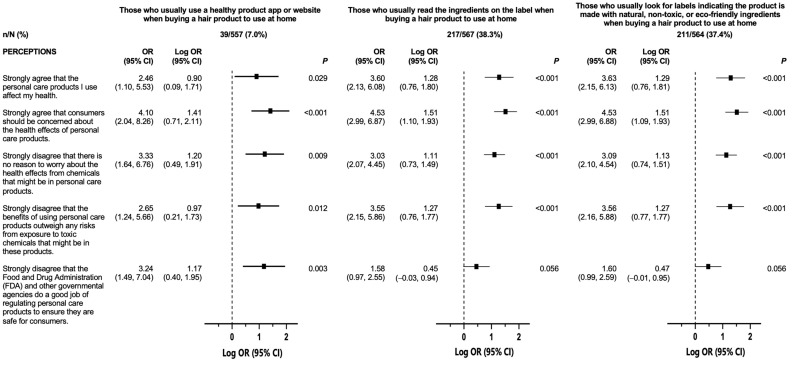
Adjusted odds ratios and 95% confidence intervals of the associations between perceptions of harm and hair-product-purchasing behaviors. Associations between perceptions of harm (strongly agreed/strongly disagreed with a perception vs. all other responses) and hair-care-purchasing behaviors were examined using multivariable-adjusted logistic regression models. Associations were reported as odds ratios and 95% confidence intervals. Each plot illustrates the log odds of always/usually exhibiting “safer” behaviors around PCP purchasing, adjusted for age (continuous), gender, income, marital status, and race and ethnicity, when compared with those who did not always/usually practice “safer” purchasing behaviors.

**Figure 3 ijerph-20-07129-f003:**
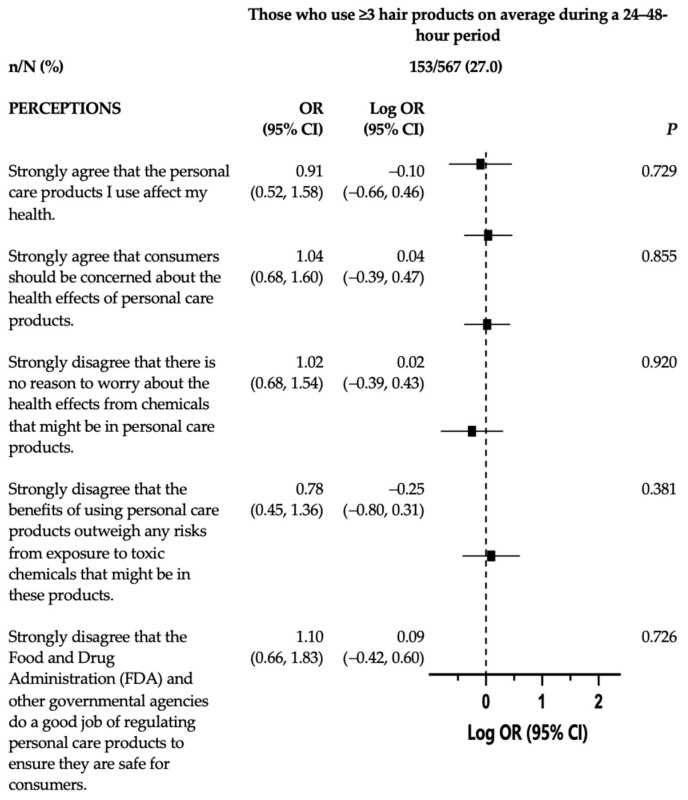
Adjusted odds ratios and 95% confidence intervals of the associations between perceptions of harm and hair product usage behaviors. Associations between perceptions of harm (strongly agreed/strongly disagreed with a perception vs. all other responses) and hair-care-use behaviors were examined using multivariable-adjusted logistic regression models. Associations were reported as odds ratios and 95% confidence intervals. Each plot illustrates the log odds of always/usually exhibiting “safer” behaviors around PCP hair product use, adjusted for age (continuous), gender, income, marital status, and race and ethnicity, when compared with those who did not always/usually practice “safer” purchasing behaviors.

**Table 1 ijerph-20-07129-t001:** Sociodemographic characteristics of study participants aged 18–79 years (*n* = 567).

		Gender Identity	
Characteristics	Overall	Females	Males
*n* (%)	567 (100.0)	485 (85.5)	82 (14.5)
Age (years), mean ± SD	37.0 ± 15.6	36.8 ± 15.2	38.6 ± 17.5
Race and ethnicity			
Asian American/Pacific Islander	114 (20.1)	52 (9.2)	2 (0.4)
Hispanic/Latinx	57 (10.1)	260 (45.9)	51 (9.0)
Non-Hispanic Black	54 (9.5)	49 (8.6)	8 (1.4)
Non-Hispanic White	311 (54.9)	97 (17.1)	17 (3.0)
Multiracial or other	31 (5.5)	27 (4.8)	4 (0.7)
Education			
High school diploma	24 (4.2)	19 (3.4)	5 (0.9)
Some college	122 (21.5)	108 (19.1)	14 (2.5)
Bachelor’s degree	182 (32.1)	161 (28.4)	21 (3.7)
Master’s degree	126 (22.2)	113 (19.9)	13 (2.3)
Doctoral degree	113 (19.9)	84 (14.8)	29 (5.1)
Income			
Less than USD 50,000	90 (15.9)	74 (13.1)	16 (2.8)
USD 50,000–99,999	171 (30.2)	152 (26.8)	19 (3.4)
USD 100,000–149,999	156 (27.5)	133 (23.5)	23 (4.1)
USD 150,000 and above	150 (26.5)	126 (22.2)	24 (4.2)
Marital Status			
Married	216 (38.1)	181 (31.9)	35 (6.2)
Unmarried	351 (61.9)	304 (53.6)	47 (8.3)
University Role			
Undergraduate student	87 (15.3)	75 (13.2)	12 (2.1)
Graduate student	139 (24.5)	117 (20.6)	22 (3.9)
Staff	228 (40.2)	209 (36.9)	19 (3.4)
Faculty	87 (15.3)	67 (11.8)	20 (3.5)
Other	26 (4.6)	17 (3.0)	9 (1.6)

Note: Proportions may not sum to 100 due to rounding.

**Table 2 ijerph-20-07129-t002:** Sociodemographic characteristics of study participants aged 18-79 years (N=567) in relation to perceptions and purchasing behavior towards personal care products.

		Perceptions					Purchasing Behavior		
Characteristics	Overall	Strongly Agree That the Personal Care Products I Use Affect My Health ^a^.	Strongly Agree that Consumers Should Be Concerned about the Health Effects of Personal Care Products ^a^.	Strongly Disagree that There Is no Reason to Worry about the Health Effects from Chemicals That Might be in Personal Care Products ^b^.	Strongly Disagree That the Benefits of Using Personal Care Products Outweigh any Risks from Exposure to Toxic Chemicals that Might be in These Products ^b^.	Strongly Disagree that the Food and Drug Administration (FDA) and Other Governmental Agencies Do a Good Job of Regulating Personal Care Products to Ensure They Are Safe for Consumers ^b^.	Those Who Usually/Always Use a Healthy Product App or Website when Buying a Hair Product to Use at Home ^c^.	Those Who Usually/Always Read the Ingredients on the Label when Buying a Hair Product to Use at Home ^c^.	Those Who Usually/Always Look for Labels Indicating the Product Is Made with Natural, Non-Toxic or Eco-Friendly Ingredients when Buying a Hair Product to Use at Home ^c^.
**N (%)**	567 (100.0)	77/567 (13.6)	153/567 (27.0)	192/567 (33.9)	86/567 (15.2)	91/567 (16.1)	39/557 (7.0)	217/567 (38.3)	211/564 (37.4)
**Gender identity**									
Female	485 (85.5)	70 (12.4)	140 (24.7)	176 (31.0)	74 (13.1)	84 (14.8)	34 (6.0)	197 (34.7)	189 (33.3)
Male	82 (14.5)	7 (1.2)	13 (2.3)	16 (2.8)	12 (2.1)	7 (1.2)	5 (0.9)	20 (3.5)	22 (3.9)
**Race and ethnicity**									
Asian American/Pacific Islander	114 (20.1)	13 (2.3)	27 (4.8)	27 (4.8)	16 (2.8)	12 (2.1)	9 (1.6)	32 (5.6)	36 (6.4)
Hispanic/Latinx	57 (10.1)	8 (1.4)	20 (3.5)	22 (3.9)	10 (1.8)	11 (1.9)	8 (1.4)	29 (5.1)	31 (5.47)
Non-Hispanic Black	54 (9.5)	6 (1.1)	20 (3.5)	31 (5.5)	12 (2.1)	8 (1.4)	7 (1.2)	36 (6.4)	32 (5.6)
Non-Hispanic White	311 (54.9)	44 (7.8)	73 (12.9)	98 (17.3)	45 (7.9)	58 (10.2)	12 (2.1)	100 (17.6)	93 (16.4)
Multiracial/Other	31 (5.5)	6 (1.1)	13 (2.3)	14 (2.5)	3 (0.5)	2 (0.4)	3 (0.5)	20 (3.5)	19 (3.4)
**Age, years**									
18-33	295 (52.0)	37 (6.5)	83 (14.6)	94 (16.6)	38 (6.7)	52 (9.2)	19 (3.4)	113 (19.9)	116 (20.5)
34-49	123 (21.7)	16 (2.8)	35 (6.2)	43 (7.6)	22 (3.9)	25 (4.4)	12 (2.1)	49 (8.6)	47 (8.3)
50-79	149 (26.3)	24 (4.2)	35 (6.2)	55 (9.7)	26 (4.6)	14 (2.5)	8 (1.4)	55 (9.7)	48 (8.5)
**Education**									
High school diploma	24 (4.2)	3 (0.5)	8 (1.4)	8 (1.4)	7 (1.2)	2 (0.4)	2 (0.4)	8 (1.4)	7 (1.2)
Some college	122 (21.5)	15 (2.7)	30 (5.3)	40 (7.1)	13 (2.3)	12 (2.1)	10 (1.8)	47 (8.3)	40 (7.1)
Bachelor’s degree	182 (32.1)	31 (5.5)	55 (9.7)	68 (12.0)	31 (5.5)	33 (5.8)	16 (2.8)	79 (13.9)	84 (14.8)
Master’s degree	126 (22.2)	20 (3.5)	38 (6.7)	47 (8.3)	21 (3.7)	26 (4.6)	8 (1.4)	50 (8.8)	52 (9.2)
Doctoral degree	113 (19.9)	8 (1.4)	22 (3.9)	29 (5.1)	14 (2.5)	18 (3.2)	3 (0.5)	33 (5.8)	28 (4.9)
**Income**									
Less than $ 50,000	90 (15.9)	10 (1.8)	28 (4.9)	34 (6.0)	17 (3.0)	21 (3.7)	8 (1.4)	36 (6.4)	40 (7.1)
$50,000–$99,999	171 (30.2)	18 (3.2)	44 (7.8)	59 (10.4)	19 (3.4)	22 (3.9)	11 (1.9)	76 (13.4)	69 (12.2)
$100,000–$149,999	156 (27.5)	29 (5.1)	47 (8.3)	56 (9.9)	28 (4.9)	25 (4.4)	12 (2.1)	61 (10.8)	60 (10.6)
$150,000 and above	150 (26.5)	20 (3.5)	34 (6.0)	43 (7.6)	22 (3.9)	23 (4.1)	8 (1.4)	44 (7.8)	42 (7.4)
**Marital Status**									
Married	216 (38.1)	32 (5.6)	52 (9.2)	67 (11.8)	34 (6.0)	30 (5.3)	20 (3.5)	80 (14.1)	79 (13.9)
Unmarried	351 (61.9)	45 (7.9)	101 (17.8)	125 (22.1)	52 (9.2)	61 (10.8)	19 (3.4)	137 (24.2)	132 (23.3)

Note: Proportions may not sum up to 100 due to rounding. ^a^ Strongly agree versus all other responses. ^b^ Strongly disagree versus all other response options. ^c^ Always/usually versus all other response options.

## Data Availability

The data presented in this study are available upon request from the corresponding author. The data are not publicly available due to privacy restrictions.

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
