# Peer review of "Beauty Beware: Associations between Perceptions of Harm and Safer Hair-Product-Purchasing Behaviors in a Cross-Sectional Study of Adults Affiliated with a University in the Northeast"

_ijerph, 2023, doi:10.3390/ijerph20237129_

Round 1

Reviewer 1 Report

Comments and Suggestions for Authors

This study by Payne et al. investigated associations between harm perceptions and hair product purchasing patterns in a large cross-sectional sample of individuals from a college campus in the Northeastern US. The paper is exceptionally well written and adds to the current knowledge on hair product purchasing patterns in U.S. adults, with important implications for chemical exposures. I have some minor comments and suggestions below.

Introduction

·        I would recommend rephrasing feminine hygiene products as menstrual products or vaginal products (or something along those lines) to be more gender inclusive.

·        Please provide a reference for the statement on Lines 57-59.

·        The introduction is very well written, but I think it’s quite long. Could the authors consider synthesizing the information in the Introduction and save more detailed descriptions of other studies for the Discussion?

Methods

·        Can the authors describe how they chose potential confounding variables?

·        It would be helpful to note in the Methods that ORs (with their specific interpretations) and 95% confidence intervals were estimated from the logistic regression models.

·        Please consider removing p-values and statistical significance language (here and in the Results) and focus results instead on the precision of the confidence intervals.

·        The authors note in Figure 1 that 2 participants (1 who identified as non-binary and 1 who identified as genderqueer) were excluded from the analysis, but this was not discussed in the Methods. Please provide a clear rationale for the exclusion of these participants.

·        Can the authors please provide rationale for performing a complete case analysis (according to Figure 1) rather than imputing missing data? I worry about the potential for selection bias by restricting to complete data (even though the authors removed a small % of participants), particularly for the exclusion of participants that did not complete the survey. Were there patterns in missingness? I.e., were participants that did not complete the survey more likely to be missing data on specific product-related questions?

Discussion

·        The Discussion is very well written! I appreciate the authors’ discussion of their findings in the context of policies, including the importance of not placing the onus on consumers to reduce exposures.

·        The authors note the potential limited generalizability of their results given the high SES of their population, which I agree with. I wonder if the authors could consider noting here that lower SES consumers may have less access to “cleaner” products that are often more expensive? It seems that investigating this research question in a lower SES population is an important direction for future research.

·        I’m concerned with the potential for dependent misclassification because both the independent and dependent variables were measured on the same survey. For example, it is plausible that participants that were more likely to report misclassified perceptions (i.e., strongly agree that products affect health) were also more likely to report misclassified levels of app use (i.e., always use); thus, the error in both variables could be correlated. This could potentially lead to a large and unpredictable bias and should be discussed as a study limitation. 

Reviewer 2 Report

Comments and Suggestions for Authors

Comments for authors

This manuscript presents the results of survey data analysis about the relationship between the perception of toxic substances contained in cosmetic hair products and the purchasing behaviour. It is shown that the consumers who are concerned with the potential harmful health effects of personal care products are also more attentive to the product purchased (read labels, check website or specialized apps). Surprisingly, no relationship is found between the potential heal effects and the number of products used.

General comments:

The introduction, method and results sections are clear. The method is well described and relies on other studies, the results are easy to understand, the tables and figures are comprehensive. Meanwhile, the results are a bit simplistic to me. Indeed, the authors make a strong focus on a small part of the data: the individuals who care about their health and therefore adapt their purchasing behavior. Nothing is said about other individuals. Is there also clear relationships among other sub-populations of the study? While reading, I also miss stratified results if possible: by age, but at least by ethnicity because this particular point is discussed further. The discussion is quite long (more or less one third of the whole manuscript if I exclude tables and figures) and the relationship between certain paragraphs and the study results are unclear. It is particularly the case of the paragraph dedicated to the FDA regulation (p12 l378 and following) and the paragraph dedicated to the eurocentricity of beauty (p12 l409 and following). In the same time, I was expecting a part of the discussion more clearly related to the relationship between the merchandising of cosmetics producers and the expectations of the consumers. I’m not a specialist in cosmetics, but I suppose that producers sometimes emphasize a lot on the fact that their product is “XXX-free” but in the same time, the product contain the substance YYY which has more or less the same potential health effects. In other words, I think it could be interesting to discuss the questions used as outcomes of the study: are the consumers concerned with their health are also mislead in their evaluation of products? Is the label clear and relevant? Is the app or website relevant, comprehensive and independent?

To conclude, I would say that the introduction and methods sections are ok as they are. They do not need any modifications. The results should be completed with (1) data about individuals who pretend to care less about their health and (2) stratified data, in particular regarding “race and ethnicity” variable. The discussion should be reduced and improved in the way that all paragraphs must clearly be in relationship with the data and/or the results.

Round 2

Reviewer 2 Report

Comments and Suggestions for Authors

Thank you for your answers to my comments!

Best regards,